# Optimization of a Method for the Concentration of Genetic Material in Bacterial and Fungal Communities on Fresh Apple Peel Surfaces

**DOI:** 10.3390/microorganisms8101480

**Published:** 2020-09-26

**Authors:** Alexis Hamilton, Scott J. Harper, Faith Critzer

**Affiliations:** 1School of Food Science, Washington State University, Pullman, WA 99164, USA; faith.critzer@wsu.edu; 2Department of Plant Pathology, Washington State University, Pullman, WA 99164, USA; scott.harper@wsu.edu

**Keywords:** apple, DNA extraction, microbiome, optimization, sample collection, surfactant

## Abstract

Apples are the most consumed fruit in the United States and have recently been shown to exhibit some vulnerability to contamination across the supply chain. It is unclear what role a fruit microbiome analysis may serve in future food safety programs interested in understanding changes in the product and the processing environment. Ultimately, sample integrity is key if any of these approaches are to be employed; low microbial loads on apple surfaces, the inability to sample the entire surface, and inefficiency of removal may act as barriers to achieving high-quality DNA. As such, the objective of this study was to identify a reproducible method to concentrate and quantify bacterial and fungal DNA from fresh apple surfaces. Five methods were evaluated: two variations of wash solutions for bath sonication, wash filtration, epidermis excision, and surface swabbing. Epidermis excision returned the highest mean DNA quantities, followed by the sonicated washes and wash filtration. Surface swabbing was consistently below the limit of detection. Based on the quantity of host DNA contamination in surface excision, the sonicated wash solution containing a surfactant presents the greatest opportunity for consistent, high-yielding DNA recovery from the entire apple surface.

## 1. Introduction

Apples are the most commonly consumed fruit in the United States, with fresh market consumption averaging 10.7 pounds per person annually [1]. Previously, they have been considered a relatively moderate risk commodity with regards to the number of foodborne outbreaks with which they have been associated [2]. However, recent outbreak and recall events have highlighted the crop’s vulnerability to contamination [3,4,5,6,7]. Microbiome-based metagenomic methods have been suggested as a modern approach to enhancing food safety through observing changes in the microbial community of a food or the food-processing environment via the emergence of foodborne pathogens in a sample or changes in overall community populations [8]. Studies of the microbial diversity of food products and food production environments have been conducted involving kimchi and other fermented vegetable products [9,10], Latin-style and artisan cheeses [11,12], and a variety of whole and cut fresh produce [13,14,15,16] as a way to describe the natural microflora and improve the quality of the product. Metagenomic methods have also been used as an approach to identify the presence of common foodborne pathogens in various products and production environments, including cabbage [17], cattle [18], cheese [19], cilantro [20], tomato [21], and spinach [22]; however, potentially, the most effective use of this approach to enhancing food safety lies in utilizing metagenomic methods to identify ways in which the microbiota of the production environment affect foodborne pathogen survival and persistence over time and across commodities [23,24,25].

In contrast to the relatively high abundance of microorganisms in the plant rhizosphere [26], microenvironments within the phyllosphere, specifically the carposphere, pose a significant barrier to high-quality microbiome research due to their low abundances of microorganisms, particularly on the fruit surface [27]. This is partially affected by conditions of the growing environment to which the aerial structures are exposed, including ultraviolet light exposure [28], limited nutrients and desiccation [29], and the physiological status of the plant [30]. Low microbial loads of target microorganisms are linked to low quantities of associated DNA, thus requiring enrichment procedures to increase microbial populations and to ensure adequate DNA quantities are available for downstream amplification and sequencing activities [31,32]. This practice may over- or underrepresent some communities through artificial selection based on incubation temperature, nutrient composition of the growth medium, incubation time, or other compounding factors [20,33,34,35]. Although the advent of next- and third-generation sequencing technologies and improved bioinformatics tools have improved the accuracy, quality, and speed of obtaining read outputs, DNA quantities of greater than 10 ng/μL are recommended [36,37,38].

DNA extraction methodologies may also introduce an opportunity for bias by selecting for some types of bacterial communities over others, preventing effective comparisons across studies using different methods [39,40,41,42]. Variations in the microbial cell structure affect the efficiency of cell lysis [41,43], but it has been shown that the addition of a bead-beating step could aid in more accurate community representation through more effective cell lysis and potentially overcome this inherent opportunity for bias [44,45]. Three primary extraction methodologies are used: phenol-chloroform, mechanical, and chemical/enzymatic [39]. It has been shown that extraction methodologies incorporating a mechanical lysis step were more likely to result in sheared DNA samples over phenol-chloroform or chemical/enzymatic methods [46]. However, variations in DNA quantity and quality are seen across kits that boast similar extraction strategies [45,47,48,49]. Additionally, it has also been suggested that the greatest source of the resulting variation attributed to extraction methodology efficiency lies more in the uncontrollable biological variations from microbiome samples over the extraction methodology itself [50], highlighting the need to select a methodology optimized to reliably isolate representative samples.

Despite the rapid advances in popularity and efficacy of the tools for microbiome research in recent years, there is still a need for a reliable method for sampling microorganisms from surfaces that are known to contain low microbial loads and, thus, low DNA concentrations. The objective of this study was to identify a reproducible sampling method capable of removing bacterial and fungal populations from fresh apple surfaces from which high-yielding DNA could be isolated without host DNA contamination.

## 2. Materials and Methods 

Fresh, unwaxed Gala samples were supplied by the Washington Tree Fruit Research Commission (Wenatchee, WA, USA) and stored at 4 °C prior to microbiome sample collection. Microbial species were collected from stored apple surfaces before undergoing DNA extraction and quantification. Five methods were evaluated by mean DNA yields: bath sonication (two variations), epidermis excision, surface swabbing, and wash filtration (Figure 1). The experimental design resulted in three independent biological replicates per treatment and three technical replicates per biological replicate (n = 9, n_total_ = 45).

### 2.1. Bath Sonication

Two bath sonication methods were utilized to assess how manipulation of the wash solution would affect DNA yields. In both methods, apples were placed in a sterile, unlined buffer bag (Fisher Scientific, Fair Lawn, NJ, USA) and submerged in 250 mL of the wash solution before sonication using the Branson 3510 Ultrasonic Cleaner (Branson Ultrasonics, Danbury, CT, USA). The sonication liquid was degassed for five minutes to remove trapped gasses and enhance the sonication efficiency. 

#### 2.1.1. Wash Solution Variation and Method 1

The apple was submerged in a wash solution containing 1X Tris-EDTA (TE) buffer (Fisher Scientific, Fair Lawn, NJ, USA). The sample bag was inserted into the cleaner basket and lid secured before being sonicated for 5 min at 40 kHz. The 1X TE solution was separated into ten 25-mL aliquots and centrifuged at 4000× *g* for 20 min at 4 °C. The supernatant was discarded, and the pellet was resuspended in 1-mL 1X TE. This was repeated for the nine remaining subsamples before pooling all subsamples together for a final centrifugation at 4000× *g* for 20 min at 4 °C. The supernatant was discarded, and the pellet was resuspended in 250 µL of 1X TE for extraction.

#### 2.1.2. Wash Solution Variation and Method 2

The second wash solution was composed of 1X TE buffer with 2% Tween 80 (Fisher Scientific, Fair Lawn, NJ, USA). The sample bag was hand-massaged for 1 min and inserted into the cleaner basket and lid secured before being sonicated for 5 min. The wash solution was separated into ten 25-mL aliquots and centrifuged at 10,000× *g* for 10 min at 4 °C. The supernatant was discarded, and the pellet was resuspended in 1-mL 1X TE. This was repeated for the nine remaining subsamples before pooling all subsamples together for a final centrifugation at 10,000× *g* for 10 min at 4 °C. The supernatant was discarded, and the pellet was resuspended in 250 µL of 1X TE for extraction.

### 2.2. Wash Filtration

Apples were placed in a sterile, unlined buffer bag, submerged in 250 mL of 1X TE solution, and hand-massaged for 1 min before sonication using an ultrasonic cleaner. The sonication liquid was degassed for five minutes. The sample bag was inserted into the cleaner basket with the stem end up and lid secured before being sonicated for 5 min at 40 kHz. The sample bag was removed, hand-massaged for 1 min, and reinserted into the cleaner basket with the calyx end up and sonicated for 5 min. The solution was then vacuum-filtered through a sterile 0.45-μm S-Pak Membrane Filter (Millipore SAS, Molsheim, France) using the Pall manifold filtration system (Pall Corporation, Port Washington, NY, USA). The filter was submerged in 25-mL 1X TE with 2% Tween 80 solution in a 50-mL centrifuge tube (Thermo Fisher Scientific, Incorporated, Waltham, MA, USA), sealed with parafilm (Bemis Company, Incorporated, Oshkosh, WI, USA) to prevent leakage or contamination, and sonicated for 5 min. The filter was discarded, and the liquid solution centrifuged at 10,000× *g* for 10 min at 4 °C. The supernatant was discarded, the pellet resuspended in 1-mL 1X TE, and the tube centrifuged again at 10,000× *g* for 10 min at 4 °C. The supernatant was discarded, and the pellet was resuspended in 250 µL of 1X TE for extraction. 

### 2.3. Epidermis Excision

A 1-cm^2^ sample of the apple epidermis was excised using a sterile #10 scalpel (Exelint International, Co., Redondo Beach, CA, USA). The skin sample was weighed and standardized to 250 ± 1 mg for extraction.

### 2.4. Surface Swabbing

A 25-cm^2^ area of the apple surface was swabbed using a sterile cotton tip swab moistened with Letheen broth (3M, St. Paul, MN, USA). The swab tip was aseptically removed to be directly processed for extraction. To prevent artificial selection of some species, no samples were enriched prior to DNA extraction and quantification.

### 2.5. Microbial DNA Extraction

DNA extraction was performed using an enzymatic kit, ZymoBIOMICS DNA Miniprep Kit (Zymo Research, Irvine, CA, USA). Samples were processed as described in the ZymoBIOMICS protocol. Prior to extraction, each Zymo-Spin^TM^ III-HRC filter was prepared by adding 600 µL of ZymoBIOMICS^TM^ HRC Prep Solution and centrifuged at 8000× *g* for 3 min at 4 °C.

Samples were placed in a ZR BashingBead^TM^ lysis tube (0.1 and 0.5 mm) to which 750-µL ZymoBIOMICS lysis solution was added. The tube cap was tightened to prevent leakage and secured in a FastPrep^®^—24-bead beater with a 2-mL tube holder assembly (MP Biomedicals, Irvine, CA, USA). The tubes were processed at maximum speed (6.0 m per second, m/s) for 1 min with a 5-min rest period to prevent overheating the sample, repeated for a total of 5 cycles. The tube was centrifuged at 10,000× *g* for 1 min at 4 °C, and 400 µL of the supernatant transferred to a Zymo-Spin^TM^ III-F filter in a collection tube. The tube was centrifuged at 8000× *g* for 1 min at 4 °C and the filter discarded. 

To facilitate DNA binding, 1200 µL of ZymoBIOMICS^TM^ DNA binding buffer was added to the filtrate in the collection tube and mixed by pipetting. An 800-µL sample of the mixture was added to a Zymo-Spin^TM^ IICR column in a new collection tube and centrifuged at 10,000× *g* for 1 min at 4 °C. The flow-through was discarded and previous step repeated with the remaining 800 µL of the mixture. The filter was transferred to a new collection tube and washed with 400 µL of ZymoBIOMICS^TM^ DNA wash buffer 1. The sample was centrifuged at 10,000× *g* for 1 min at 4 °C and the flow-through discarded. The filter was transferred to a new collection tube and washed a second time with 700 µL of ZymoBIOMICS^TM^ DNA wash buffer 2. The sample was again centrifuged at 10,000× *g* for 1 min at 4 °C and the flow-through discarded. The sample was washed a third time with 200 µL of ZymoBIOMICS^TM^ DNA wash buffer 2 and centrifuged at 10,000× *g* for 1 min at 4 °C. 

The column was transferred to a sterile 2-mL microcentrifuge tube and 100 µL of ZymoBIOMICS^TM^ DNase/RNase free water added directly to the column and incubated for one minute. The sample was then centrifuged at 10,000× *g* for 1 min at 4 °C to elute the DNA. The DNA sample was transferred to a prepared Zymo-Spin^TM^ III-HRC filter in a new, sterile 2-mL microcentrifuge tube and centrifuged at exactly 16,000× *g* for 3 min at 4 °C, per explicit manufacturer instructions. The filter was discarded, and extracted DNA was stored at 4 °C until DNA quantification.

### 2.6. Quantification of Genetic Material

Genomic DNA was quantified with the Qubit 4.0 Fluorometer (Thermo Fisher Scientific, Incorporated, Waltham, MA, USA) per the manufacturer’s instructions. The double-stranded (ds)DNA High Sensitivity assay was calibrated using the provided reference standards immediately prior to sample quantification. A 5-μL sample of DNA was added to 195-μL newly prepared working solution (composed of a 200-fold dilution of fluorescent dye in sterile buffer), vortexed, and incubated at room temperature for 2 min prior to reading. 

### 2.7. Quantitative Real-Time Polymerase Chain Reaction (qPCR) of Host, Bacterial, and Fungal Genes

To quantify the yield of plant, bacterial, and fungal DNA obtained through all five methods, a qPCR assay was utilized. 

#### 2.7.1. Preparation of Microbiome Samples

A second set of fresh, unwaxed Gala samples were processed using the methods described above. Additionally, culturable bacterial and fungal load were determined for each method via serial dilution in 0.1% peptone and plating in duplicate on tryptic soy agar (TSA; Becton, Dickinson and Company, Sparksville, MD, USA) and potato dextrose agar (PDA; Becton, Dickinson and Company, Sparksville, MD, USA) for bacterial and fungal enumeration, respectively. TSA plates were incubated at 35 °C for 2 days and PDA for 25 °C for 5 days prior to enumeration to estimate log_10_ colony-forming units (CFU) per sample. Microbiome samples were collected in triplicate by an extraction method (n = 3, n_total_ = 15) utilizing both the microbial and plant DNA extraction methodologies. Microbiome samples were processed as previously described. 

Along with microbiome extraction, samples for the plant DNA extraction were processed in parallel using the column-based DNeasy Plant Mini Kit (QIAGEN, Germantown, MD, USA). Briefly, liquid and solid samples were stored at 4 °C and −80 °C, respectively, until extraction. Solid samples were disrupted using the TissueLyser II (QIAGEN, Germantown, MD, USA) for 1 min at a frequency of 20 Hz/s before the remaining extraction steps. Manufacturer’s instructions were followed for processing all samples, and isolated DNA was stored at −20 °C until qPCR.

#### 2.7.2. Preparation of Bacterial, Fungal, and Plant Standards

The following isolates were utilized as standards: *Escherichia coli* K12 (bacterial), *Penicillium expansum* (fungal), and an excised section of apple epidermis, *Malus domestica* “Gala” (plant). A pure culture of *E. coli* was incubated overnight in tryptic soy broth (TSB; Becton, Dickinson and Company, Sparksville, MD, USA) at 35 °C, and a 250-μL sample was stored at 4 °C until DNA extraction. A pure culture of *P. expansum* was obtained via incubation on PDA for 7 days at 25 °C. A 10-mL sample of sterile 0.1% peptone was pipetted onto the plate surface, and a sterile spreader (Fisher Scientific, Fair Lawn, NJ, USA) was used to gently scrape the surface to release the spores from the conidiophore. From this suspension, a 250-μL sample was obtained and stored at 4 °C until DNA extraction. Lastly, for the plant standard, a 200-mg sample was aseptically removed from the equator of a Gala apple using a sterile #10 scalpel. This sample was stored at −80 °C until DNA extraction. DNA was extracted from the bacterial and fungal standards via the microbial extraction method, while the plant standard underwent DNA extraction via the plant extraction method, as described above.

Standards were amplified through conventional PCR using the primer sets and cycle information described in Table 1. A 20-μL reaction was prepared for each standard in quintuplicate with the following parameters: 10–100-ng DNA template, 500-nM forward primer, 500-nM reverse primer, 6-μL nuclease-free water, and 10-μL PowerUp™ SYBR™ Green Master Mix (2X) (Thermo Fisher Scientific, Incorporated, Waltham, MA, USA). PCR products were purified using the column-based GenElute PCR Clean-Up Kit (QIAGEN, Germantown, MD, USA) following the manufacturer’s instructions. Purified products were quantified using the Qubit 4.0 fluorometer to calculate the copy number, and samples were stored at −20 °C until qPCR. 

#### 2.7.3. qPCR Design

Purified standards were serially diluted in nuclease-free water prior to amplification. All samples were analyzed in triplicate on a CFX96 Touch Real-Time PCR Detection System (Bio-Rad, Hercules, CA, USA) using the reaction conditions described above and in Table 1. No template controls were included for every assay as a negative control. A standard curve was constructed by plotting the quantification cycle (C_q_) for the standard and each subsequent 10-fold dilution against the natural log of the copy number (copies/μL), from which sample C_q_ values were determined.

### 2.8. Statistical Analysis

Data were analyzed using SAS University Edition (SAS Institute, Cary, NC, USA). One-way Analysis of Variance (ANOVA) and Tukey honest significant differences (HSD) tests were performed using the general linear mixed model (GLIMMIX) procedure to assess the presence of and differences between microbiome collection methods and DNA yields (*p* ≤ 0.05). Samples that were below the limit of detection were assigned a value of 0.4 pg/μL for statistical analysis. 

## 3. Results

Epidermis excision provided the highest mean quantity of DNA, followed by wash solution #2, wash solution #1, wash filtration, and surface swabbing. Mean quantities of DNA are reported in Table 2. We found that the sample collection methods recovered significantly different quantities of DNA (one-way ANOVA, F_4,40_ = 4.84, *p* = 0.0039). Tukey’s post-hoc mean separation showed significant differences between the epidermis excision and wash filtration and surface swabbing (Table 2). 

Except for epidermis excision, at least one sample in each method was not able to recover DNA from the apple peel in quantities above the limit of detection for the fluorometer (0.5 pg/μL). Of all the remaining methods, wash solution #2 most consistently recovered DNA above the limit of detection for 89.9% of the samples analyzed.

Despite the variations in DNA recovery, the bacterial and fungal populations were consistently recovered across all methods, with the population size varying by the quantity of apple surface sampled (Table 3). 

Bacterial and fungal populations were both recovered in significantly greater quantities from the methods that sampled the whole apple: wash solution #1, wash solution #2, and wash filtration (*p* ≤ 0.05). Among the whole-apple samples, there were no significant differences in fungal population recovery; however, wash solution #1 recovered significantly less bacterial populations than wash solution #2 and wash filtration (*p* ≤ 0.05).

When quantifying the target copy number with qPCR, the bacterial and fungal targets were present in the microbiome samples at quantities above the limit of detection in all sample methodologies (Table 4). The only methodology to contain quantifiable plant DNA was the epidermis excision method. Additionally, fungal targets were present in larger quantities than bacterial targets across all methods evaluated.

There were no significant differences between *16S* gene target quantities across the methods; however, there were statistically significant differences in fungal target amplifications. The wash filtration method recovered the greatest quantity of fungal targets, followed by wash solution #2 and surface swabbing, which were all significantly greater than those recovered using the wash solution #1 or epidermis excision methods. Plant DNA recovery was significantly greater in the epidermis excision method than in all other methods, since this was the only methodology for which a plant target was amplified. 

## 4. Discussion

Microbial loads recovered from apple surfaces in this study were up to 3-log higher than those reported in a previous study quantifying bacterial and/or fungal loads on fresh apples as recovered via different washing solutions [55]. Sare et al. (2020) reported that wash solutions containing a surfactant component, incorporating a sonication step, and utilizing multiple wash steps increased the quantities of bacteria recovered in the wash solution [55], which was similarly reflected in this study across the sonication and filtration methods evaluated. Additionally, the quantity of fungal DNA present compared to bacterial observed in this study was contrary to that reported in previous studies that found greater quantities of prokaryotic to eukaryotic DNA [56]. This finding may be affected by the length of time the apples were in storage prior to analysis. The apples used in this study were in storage for up to seven months prior to extraction, and fungal populations during storage have been shown to increase at a higher rate of growth relative to bacterial populations [57,58,59].

While the epidermis excision method returned the highest genomic DNA yield and consistently recovered DNA at quantities above the limit of detection (due to processing the peel surface directly), this method also was the only one evaluated to show the potential for host DNA contamination in the sample (Table 4). The problem of nontarget DNA contamination is not new to microbiome research and has been previously reported in early studies of the human gut microbiome, in which it is often necessary to directly sample intestinal surfaces to accurately quantify and describe the related microbial community. Specifically, Chiodini et al. (2015) reported an inability to accurately enumerate the microbial load from submucosal intestinal tissues, which was attributed to competing issues of small bacterial populations and large quantities of human host cell contamination [60]. This led to a subsequent study to quantify the amount of host (human) contamination present using this direct processing method, which was estimated to be as high as 75% across the samples [61]. 

It should be noted that a plant DNA extraction methodology was used to process all samples in the series with the microbial extraction method. This would give any samples the best possible chance of yielding host DNA if present. In a standard microbial DNA extraction procedure, cells are lysed using enzymatic, thermal, and/or mechanical methods, and the DNA is separated from other cell components or debris through a series of washing steps, all while maintaining genomic integrity through to the final elution step. Conversely, plant DNA extraction requires more mechanically- and chemically-intensive methodologies to overcome the many additional barriers to effective lysis, including the rigidity of the cell wall and the presence of inhibitory secondary metabolites [62,63,64,65]. While plant extraction kits can isolate amplifiable microbial DNA [66], methodologies targeting microbial species alone have been shown to be ineffective at optimally isolating quantifiable plant DNA [67]. Nonetheless, it is important to minimize host contamination whenever possible in upfront sample processing.

The epidermis excision method also was restricted to a small surface area of the apple (approximately 1 cm^2^) based on input thresholds dictated by the extraction kit used in these experiments, reducing the likelihood of recovering a DNA sample that was representative of the peel microbiome. This downfall could be mitigated by subsampling each apple at several key areas (equator, shoulder, stem, and calyx) [27,59], which would quadruple costs associated with extraction, as each subsample would require individual processing. 

The use of a sonicated wash solution was highly desirable due to its ability to collect a microbiome sample from the entirety of the fruit surface, increasing the likelihood of collecting a representative sample. The initial wash solution used only a buffer solution, which showed low DNA and culturable microbial recovery. The addition of a surfactant to the wash solution increased the mean DNA recovery in the wash solution by nearly 25 times. Surfactants have been used to facilitate the removal of microorganisms from fresh produce surfaces, primarily due to their ability to reduce the surface tension of the wash solution and enhance the spread across a surface [68]. Specifically, Huang et al. (2018) evaluated the influence of surfactants on a detachment of *Escherichia coli* O157:H7 on lettuce leaves, a food which, it has been hypothesized, has been commonly implicated in foodborne outbreaks and recall events in part due to its complex surface topography to enable microbial attachment [69]. The study found that the use of 0.05% and 0.1% Tween 20 in the wash water solution facilitated bacterial removal by up to 2.13-log CFU/cm^2^ across the leaf surface, with log reduction increasing with the surfactant concentration [70]. However, Pietrysiak et al. (2020) examined the effect of adding surfactants to sanitizer solutions for the removal of *Listeria innocua* from fresh apple surfaces and found that, while the addition of 0.1% Tween 20 to a peracetic acid wash solution resulted in a nearly 4-log reduction in microbial populations, this reduction was not significantly different from the removal ability of water alone, possibly due to the lower concentration of surfactant used compared to this study [71].

It has also been suggested that ultrasonication (frequencies of greater than 20 kHz) may result in cell lysis, thereby releasing free genetic material into the sonicated solution that would not be dense enough to effectively pellet [72,73]. Specifically, Starke et al. (2019) reported that ultrasonication for 10 min resulted in an unequal lysis of Gram-negative over Gram-positive or fungal cells, which would underrepresent less-resistant bacterial populations [72]. Comparatively, a study evaluating sonication in two different lysis buffers as methods for downstream extraction activities found that sonication for 20 min was insufficient to lyse cyanobacteria and release genetic material into the matrix [73]. These conflicting results may be due to differences in methodology, including equipment, frequency, sonication solution, and time of exposure [74,75]; furthermore, the method employed in this study, and particularly in the sonication wash treatments, did not exceed the conditions reported to induce bias via premature cell lysis into the sample prepared prior to DNA extraction. 

Both the wash filtration and surface swabbing methods consistently returned relatively low quantities of DNA. While filtering the wash solution was able to concentrate the sample collected during sonication, it did not increase the DNA yield. This suggests that the makeup of the wash solution itself was imperative to the collection of a high-quality sample used for subsequent processing. Given the environment of the apple surface and potential for bias or contamination of samples even by laboratory reagents [61], it is imperative to utilize a wash solution designed to maintain the optimum pH and protect DNA from degradation by inactivating harmful DNases prior to downstream analysis, for which TE buffer has been readily used [76,77]. 

Lastly, the surface swab method was consistently unable to recover sufficient quantities of DNA to be detected by the fluorometer (≥0.5 pg/μL), despite directly processing the swab itself. Although this method was quick, the poor recovery observed using this method suggests that mechanical action alone is insufficient for the removal of low numbers of microorganisms from fresh fruit surfaces. Surface swabbing is a method primarily used in the evaluation of microbial loads on food contact surfaces and does not consistently recover microorganisms when present at low concentrations and across product types [78]. A recent survey evaluating the preference and efficacy of friction-based surface swabs compared to other contact methods showed no difference in their ability to remove bacterial populations from food-processing plant surfaces, which also suggests that friction alone is unable to yield better results than low-contact alternatives, even when populations are highly concentrated [79]. This method may be improved upon by a hurdle approach, as evaluated by Branck et al. (2017), who described the efficacy of traditional surface swabbing compared to a sonicated swab in *L. monocytogenes* biofilms on stainless steel coupons [80]. The study showed higher total viable counts and a greater consistency of results when biofilms were sampled using the sonicated alternative [80]. The effect of a hurdle approach on increasing the recovery of microorganisms from fresh produce surfaces was not expressly evaluated in this study but should be the subject of future studies, as it may prove to further optimize the recovery of target organisms from samples known to contain low microbial biomass and of such size that a swab method would be preferred.

## 5. Conclusions

Based on these findings, the second wash solution containing 1X TE buffer with 2% Tween 80 provides the greatest opportunity for consistently recovering high quantities of microbial DNA from fresh apple peel surfaces. While apples are known to contain relatively low populations of microorganisms, this methodology allowed for sampling of the entire apple surface, while minimizing the risk of host DNA contamination. Future studies are needed to determine if this methodology can be easily applied to other sample types and are warranted as metagenomic studies increase in these sample types.

## Figures and Tables

**Figure 1 microorganisms-08-01480-f001:**
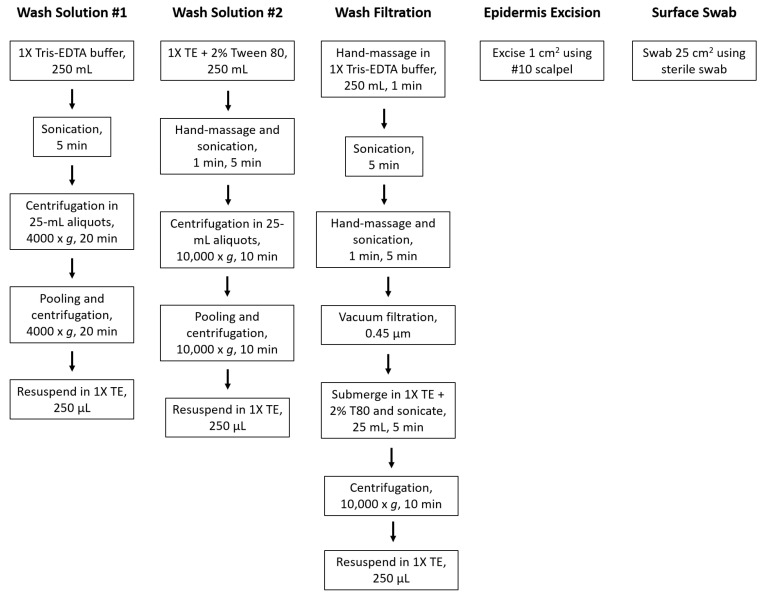
Simplified laboratory activities used for sample preparation. Activities described followed by DNA extraction and quantification.

**Table 1 microorganisms-08-01480-t001:** Primer sequences used for amplification and quantification of bacterial, fungal, and plant DNA targets.

Primer	Primer Sequence	Primer Target (Product Size; bp)	Cycle Information	Source
343F	TACGGRAGGCAGCAG	Bacterial *16S* (510)	(95 °C, 30 s; 60 °C, 60 s) × 40	[51,52]
784R	TACCAGGGTATCTAATCCT			
ITS86F	GTGAATCATCGAATCTTTGAA	Fungal *ITS2* (369)	(95 °C, 30 s; 55 °C, 30 s; 72 °C, 60 s) × 40	[53]
ITS4R	TCCTCCGCTTATTGATATGC			
ITS-p5F	CCTTATCAYTTAGAGGAAGGAG	Plant *ITS1* (408)	(94 °C, 30 s; 55 °C, 40 s; 72 °C, 60 s) × 34	[54]
ITS-u2R	GCGTTCAAAGAYTCGATGRTTC			

**Table 2 microorganisms-08-01480-t002:** Mean DNA recovery from each apple peel microbiome sample collection method.

Sample Collection Method	Mean DNA ± Std. Error (pg/μL)	Samples Below Limit of Detection (%)
Wash solution #1	24.93 ± 13.6 AB ^1^	6 (66.7)
Wash solution #2	614.82 ± 366.8 AB	1 (11.1)
Wash filtration	5.33 ± 4.9 A	8 (88.9)
Epidermis excision	661.67 ± 153.2 B	0 (0)
Surface swabbing	≤0.50 A	9 (100)

^1^ Quantities followed by different letters are significantly different (*p* ≤ 0.05).

**Table 3 microorganisms-08-01480-t003:** Mean culturable bacterial and fungal populations by sample collection method.

Sample Collection Method	Mean Microbial Populations ± Std. Dev. (Log_10_ CFU/Sample)	
	Bacterial	Fungal
Wash solution #1	4.22 ± 0.15 A ^1^	5.39 ± 0.20 A
Wash solution #2	5.24 ± 0.54 B	5.11 ± 0.39 A
Wash filtration	4.86 ± 0.35 B	5.18 ± 0.26 A
Epidermis excision	2.25 ± 0.42 C	2.00 ± 0.00 B
Surface swabbing	1.22 ± 0.28 D	2.67 ± 0.22 C

^1^ Quantities followed by different letters within a column are significantly different (*p* ≤ 0.05). CFU: colony-forming unit.

**Table 4 microorganisms-08-01480-t004:** Quantities of gene targets detected during quantitative (q)PCR by collection method.

Sample Collection Method	Estimated Log_10_ Copy Number of Gene Target ± Std. Dev.		
	Bacterial	Fungal	Plant
Wash solution #1	1.90 ± 0.00 A ^1^	5.30 ± 0.82 A	ND ^2^ A
Wash solution #2	3.04 ± 0.77 A	7.79 ± 0.20 B	ND A
Wash filtration	2.15 ± 0.83 A	8.61 ± 0.30 C	ND A
Epidermis excision	2.77 ± 0.82 A	5.29 ± 0.25 A	6.25 ± 1.23 B
Surface swabbing	1.94 ± 0.66 A	6.71 ± 0.69 D	ND A

^1^ Quantities followed by different letters within a column are significantly different (*p* ≤ 0.05). ^2^ Not detectable; gene target below the threshold of amplification.

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
