# Peer review of "Optimization of a Method for the Concentration of Genetic Material in Bacterial and Fungal Communities on Fresh Apple Peel Surfaces"

_microorganisms, 2020, doi:10.3390/microorganisms8101480_

Round 1
Reviewer 1 Report
The authors have addressed the issues which I mentioned. Thanks.
Author Response
Point 1:The authors have addressed the issues which I mentioned. Thanks.
Response 1: Thank you for reviewing our revised manuscript.
Reviewer 2 Report
General comment:
In this study, Hamilton et al. compared different DNA extraction methods to identify a reproducible and efficient method to remove fungal and bacterial DNA from apples while minimizing host DNA contamination- an important experimental step in microbiome-based approaches for microbial community estimation. A total of 5 sampling strategies- bath sonication with two different wash solutions, epidermis excision, surface swabbing, and wash filtration were compared. The authors showed that the sonication method using wash solution II retrieved the maximum amount of microbial DNA with less host DNA contamination. The current study is interesting and I appreciate the authors’ effort addressing one of the major issues in microbiome research. It will further help in determining the best postharvest management practices of apples.
It is known that bacterial and fungal communities of apple calyx, stem, and peel tissues vary depending on the storage time; and tissue type is a significant factor determining the bacterial/fungal diversity and composition. Though the current study does not aim to evaluate the temporal variation, I wonder why the calyx tissue was not considered here.
Specific comments:
Lines 19-22: It is clear from lines 19-21 that the epidermis excision and sonication method obtained the maximum mean DNA concentration amongst all. It was repeated in the following lines 21-22. I understand that authors want to reemphasize the methods that yield maximum DNA. However, please consider revising the sentence.
Lines 54-56: It is true that a sufficient quantity of DNA is a prerequisite for any generation sequencing platforms (Sanger to next-generation sequencing). Library preparation methods such as TrueSeq Nano used in HiSeq platforms require only a minimum DNA quantity of 10 ng/ul.
Lines 70-72: Further to the above comment, please remove the value <20ng/ul.
Line 121-123: Out of curiosity, is it enough to sample the epidermis of 1cm3 from one part of the apple?
Line 129: Why Zymo kit was preferred over DNeasy Power Soil Kit? Later one is known for efficient microbiome DNA extraction.
Line 276: Spelling mistake ‘yielding’
Line 313: Synonyms are nice if used properly, especially in scientific writing. Word “expressly” could be removed.
Author Response
Point 1: General comment: In this study, Hamilton et al. compared different DNA extraction methods to identify a reproducible and efficient method to remove fungal and bacterial DNA from apples while minimizing host DNA contamination- an important experimental step in microbiome-based approaches for microbial community estimation. A total of 5 sampling strategies- bath sonication with two different wash solutions, epidermis excision, surface swabbing, and wash filtration were compared. The authors showed that the sonication method using wash solution II retrieved the maximum amount of microbial DNA with less host DNA contamination. The current study is interesting and I appreciate the authors’ effort addressing one of the major issues in microbiome research. It will further help in determining the best postharvest management practices of apples. It is known that bacterial and fungal communities of apple calyx, stem, and peel tissues vary depending on the storage time; and tissue type is a significant factor determining the bacterial/fungal diversity and composition. Though the current study does not aim to evaluate the temporal variation, I wonder why the calyx tissue was not considered here.
Response 1: The calyx tissue was not considered as a source of organisms for this study because the methodologies evaluated were designed to collect organisms from the surfaces of fresh apples which are commonly consumed fresh. Since the calyx tissue is not often eaten as part of a fresh, ready-to-eat apple commodity, it was subsequently excluded from study.
Point 2: Specific comments: Lines 19-22: It is clear from lines 19-21 that the epidermis excision and sonication method obtained the maximum mean DNA concentration amongst all. It was repeated in the following lines 21-22. I understand that authors want to reemphasize the methods that yield maximum DNA. However, please consider revising the sentence.
Response 2: The second sentence (Lines 21-22), which repeated information presented in the first (Lines 19-21), was deleted and the transition in the following sentence (Lines 22-25) eliminated to reflect the recommendation.
Point 3: Lines 54-56: It is true that a sufficient quantity of DNA is a prerequisite for any generation sequencing platforms (Sanger to next-generation sequencing). Library preparation methods such as TrueSeq Nano used in HiSeq platforms require only a minimum DNA quantity of 10 ng/ul.
Response 3: Minimum DNA quantity changed to reflect the 10 ng/μl.
Point 4: Lines 70-72: Further to the above comment, please remove the value <20ng/ul.
Response 4: Addressed above.
Point 5: Line 121-123: Out of curiosity, is it enough to sample the epidermis of 1cm3 from one part of the apple?
Response 5: The authors did not think the area excised for the surface excision method was sufficient to collect a representative sample of the microbiome on the apple surface and mentioned this in the results and discussion sections (Lines 232-235, 287-292), including mean comparison analysis which showed significantly lower bacterial and fungal mean populations compared to other methods; however, the amount of the surface sampled was dictated by a maximum sample input weight by the extraction kit manufacturer.
Point 6: Line 129: Why Zymo kit was preferred over DNeasy Power Soil Kit? Later one is known for efficient microbiome DNA extraction.
Response 6: The ZymoBIOMICS DNA Miniprep Kit was determined to be a more cost-effective, consistent, and unbiased extraction procedure regarding food safety-relevant microorganisms at low bioburden for this sample type based upon our discussions with colleagues at the FDA.
Point 7: Line 276: Spelling mistake ‘yielding’
Response 7: This has been corrected.
Point 8: Line 313: Synonyms are nice if used properly, especially in scientific writing. Word “expressly” could be removed.
Response 8: This word has been removed from the manuscript.
This manuscript is a resubmission of an earlier submission. The following is a list of the peer review reports and author responses from that submission.
Round 1
Reviewer 1 Report
The publication describes the comparison of different recovery protocols to analyse the fungal and bacterial communities from fresh apple surface.
The topic is of interest in the current era of phytobiome study: how could we optimize the yield in microorganisms recovery from fruit surface ? The idea of comparing different protocols is valuable.
A scientific paper has been recently published in Microorganisms focusing on the same question: https://www.mdpi.com/2076-2607/8/3/342. This publication and its conclusion should have been mentioned in the manuscript.
There is no state-of-the art on existing protocols to recover microorganisms from apple fruits although many protocols have been published (and recently reviewed in the above mentioned publication).
For high throughput sequencing of PCR amplicon, the starting quantity of DNA can be lower than 50 ng/µl
The experimental design is relevant with 3 biological replicates and 3 technical replicates.
Nevertheless, the results are not convincing at all. Indeed, the complete experiment has been analyzed only by quantifying the DNA extracted without any additional evaluation, which far too limited. The yield of DNA is not an appropriate measure to quantify the fungi and bacteria recovered from the apple. There is no data about the proportion of plant DNA in the mix while it is a key element. The authors supposed that a protocol recover more plant DNA (which is obvious in the case of peeling) but without bringing any data supporting it.
The study did not address key questions that could bring valuable information :
- what is the yield of fungal and bacterial cell recovery (a simple plating experiment could have brought useful information on the yield in culturable microorganisms which can give already an approximation) ?
- what is the proportion of plant DNA in the extracted DNA with the 5 protocols ?
- the fungal and bacterial communities recovered are they similar with the different protocols or each protocol extract different communities ?
In conclusion, even though a protocol comparison worth testing, additional analyses, at least the high throughput sequencing of the extracted DNA (including proper controls), must be carried out to generate relevant dataset sustaining the comparison
Reviewer 2 Report
The presentation for the manuscript was clear. For the five methods, the author only measured the quantity of microbial DNA. How about the quality? The authors should also measure the quality of DNA using gel and nanodrop methods. Furthermore, the author should check the quality using at least qPCR of 16S rDNA for bacteria and ITS for fungi.